# Online Learning versus Hands-On Learning of Basic Ocular Ultrasound Skills: A Randomized Controlled Non-Inferiority Trial

**DOI:** 10.3390/medicina58070960

**Published:** 2022-07-20

**Authors:** Soo-Yeon Kang, Jonghoon Yoo, Sookyung Park, Ik-Joon Jo, Seonwoo Kim, Hyun Cho, Guntak Lee, Jong-Eun Park, Taerim Kim, Se-Uk Lee, Sung-Yeon Hwang, Won-Chul Cha, Tae-Gun Shin, Hee Yoon

**Affiliations:** 1Department of Emergency Medicine, Samsung Medical Center, Sungkyunkwan University School of Medicine, Seoul 06351, Korea; ifely@naver.com (S.-Y.K.); ikjoon.jo@samsung.com (I.-J.J.); guntak.lee@samsung.com (G.L.); jongeun7.park@samsung.com (J.-E.P.); taerimi.kim@samsung.com (T.K.); seuk.lee@samsung.com (S.-U.L.); sygood.hwang@samsung.com (S.-Y.H.); wc.cha@samsung.com (W.-C.C.); taegun.shin@samsung.com (T.-G.S.); 2Department of Emergency Medicine, College of Medicine, Kangwon National University, Chuncheon 24341, Korea; 3Department of Emergency Medicine, Keimyung University Dongsan Hospital, Daegu 41931, Korea; wsnatz@gmail.com; 4School of Nursing, University of Virginia, Charlottesville, VA 22903, USA; sookyung731.park@samsung.com; 5Biomedical Statistics Center, Research Institute for Future Medicine, Samsung Medical Center, Seoul 06351, Korea; swkimid12@naver.com; 6Biomedical Statistics Center, Samsung Medical Center, Seoul 06351, Korea; hilori@naver.com

**Keywords:** ocular ultrasound, point-of-care ultrasound (POCUS), emergency medicine, medical education, online education

## Abstract

*Background and objectives:* Ocular ultrasound is a core application of point-of-care ultrasound (POCUS) to assist physicians in promptly identifying various ocular diseases at the bedside; however, hands-on POCUS training is challenging during a pandemic. *Materials and Methods:* A randomized controlled non-inferiority trial was conducted in an academic emergency department from October 2020 to April 2021. Thirty-two participants were randomly assigned to one of two groups. Group H (hands-on learning group) participated individually in a hands-on session with a standardized patient for 30 min, whereas Group O (online learning group) learned training materials and video clips for 20 min. They scanned four eyeballs of two standardized patients sequentially following the ocular POCUS scan protocol. Repeated POCUS scans were performed 2 weeks later to assess skill maintenance. Both groups completed the pre- and post-surveys and knowledge tests. Two emergency medicine faculty members blindly evaluated the data and assigned a score of 0–25. The primary endpoint was the initial total score of scan quality evaluated using non-inferiority analysis (generalized estimating equation). The secondary endpoints were total scores for scan quality after 2 weeks, scan time, and knowledge test scores. *Results:* The least squares means of the total scores were 21.7 (0.35) for Group O and 21.3 (0.25) for Group H, and the lower bound of the 95% confidence interval (CI) was greater than the non-inferiority margin of minus 2 (95% CI: −0.48–1.17). The second scan scores were not significantly different from those of the first scan. The groups did not differ in scanning time or knowledge test results; however, Group H showed higher subjective satisfaction with the training method (*p* < 0.001). *Conclusion:* This study showed that basic online ocular ultrasound education was not inferior to hands-on education, suggesting that it could be a useful educational approach in the pandemic era.

## 1. Introduction

Point-of-care ultrasound (POCUS) is a useful tool for assisting clinicians in making appropriate decisions regarding patient evaluation and treatment at the bedside [1]. Ocular POCUS, in particular, is easy to perform because of the superficial location of the eye, and it can provide valuable diagnostic information on various conditions, including retinal detachment, vitreous hemorrhage, intraocular foreign bodies [2,3,4], and increased intracranial pressure via optic nerve sheath diameter (ONSD) measurements [5,6]. Therefore, emergency physicians (EPs) can reliably distinguish between pathologies that require urgent ophthalmologic consultation and those that do not [7,8,9]. The American College of Emergency Physicians (ACEP) recommends ocular ultrasound as one of the primary applications of ultrasound guidelines [10].

As POCUS is increasingly being used in clinical settings, ultrasound education is expanding in residency and medical student training. Traditional POCUS education consists of didactic learning accompanied by hands-on sessions. However, in a pandemic situation, such as that triggered by coronavirus disease 2019 (COVID-19), face-to-face teaching for several hours in a limited space poses an infection risk to participants, making POCUS training difficult [11,12,13]. Under these circumstances, online learning is considered to be an effective substitute, as it imposes no restrictions on the number of participants or access to teaching materials at any time or place [14,15]. The popularity of online learning in medical education has increased in recent decades [16], and the efficacy of online ultrasound training is also being investigated [16,17,18]. However, no study has evaluated the effects of online training on ocular POCUS scanning skills. Therefore, this study aimed to assess whether online ocular ultrasound training was as effective as hands-on training. Additionally, we aimed to evaluate the practicality of online ocular POCUS training using a cognitive survey following training and scan skill maintenance evaluation after two weeks.

## 2. Materials and Methods

### 2.1. Study Design

A randomized controlled non-inferiority trial was conducted in the emergency department (ED) of a tertiary academic medical center in South Korea from October 2020 to April 2021. This study was approved by the Institutional Review Board of the Samsung Medical Center (IRB file number 2020-09-174-002, accepted on 8 December 2020) and registered with ClinicalTrials.gov (NCT04834700).

### 2.2. Participants

This study required two types of participants: 3 participants serving as standardized patients (SPs) and 32 who would receive ocular POCUS scan training. SPs were recruited from adults over 18 years with no prior ophthalmic history and no specific findings on ocular ultrasonography. Among the three SPs, one participated in hands-on education, and two participated in testing after ocular POCUS education. For the ocular POCUS education, to minimize skill-dependent bias, we recruited residents and interns who worked in the emergency department but who had received no training in ocular ultrasound. Each participant and SP provided written informed consent prior to registration.

### 2.3. Study Protocol

The participants were randomized by drawing a card in a 1:1 ratio and were allocated to either the online learning group (Group O) or the hands-on learning group (Group H). Group O was provided with learning materials and video clips (approximately 20 min) for ocular POCUS scan education. Group H attended an individual 30-min hands-on session, including a lecture and ocular ultrasound scan for SP. After the session, both groups completed a pre-survey and provided their demographic data. Ocular POCUS scans were then performed sequentially on the four eyeballs of the two SPs in a fixed order, and the images were saved following a pre-distributed scan protocol. After completing the scan, all participants completed the knowledge tests and a post-survey (Table A1). The time required to complete the ocular POCUS scan in each eye was measured. Two weeks later, all participants underwent the same ocular POCUS scans on the same SPs, and the time taken to complete the scan was re-measured. Figure 1 summarizes the flow of this study.

### 2.4. Ocular POCUS

The learning materials were created based on a 2-month literature and textbook review by two emergency medicine (EM) faculty members serving as ultrasound instructors [8,19,20]. It contained sonographic findings of normal ocular anatomy and of common ophthalmic diseases such as retinal detachment and lens dislocation. In addition, we protocoled how to perform ocular POCUS scans and measure ONSD (Figure 2 and Appendix A). The participants began by selecting a predesignated ocular preset and adjusting the depth and gain to obtain an appropriate ocular scan. They then scanned and stored the ocular images of the SPs consecutively in the transverse and sagittal planes and the ONSD measurements. Ocular POCUS was performed using a Samsung ultrasound HM70A with a 7–16 MHz linear transducer (Samsung Healthcare, Seoul, Korea).

Two EM faculty members evaluated the data in a blinded manner. They assessed whether participants adjusted depth and gain adequately and acquired the entire structure from the anterior chamber to the optic nerve. They assigned scores ranging from 0 to 3 for the ultrasound setting and each of the basic views (mid-eye, tilting, and all four directions in the transverse and sagittal planes) and 0 to 4 for ONSD measurements. The scan score, ONSD score, and combined total score for each ocular scan were calculated using the average of the two expert judgments. Each ocular scan received a maximum of 25 points. As a reference value, the pre-scanned ONSD value of the experts was used. Table A2 provides a detailed description of the evaluation criteria.

### 2.5. Data Collection

The data collected for each group included age, sex, grade, number of prior ocular ultrasound scans, frequency of POCUS use, and confidence in the use of ultrasound. In addition, the scan score, ONSD score, total score for each ocular scan, ONSD value (mm) for each eyeball, and time taken to complete the scan were noted. The scores on the knowledge tests and answers to the post-survey were also collected, and some questions were answered on a five-point Likert scale.

### 2.6. Outcomes

The primary endpoint was the total score of the first ocular POCUS scan quality. The secondary endpoints included the scan and ONSD scores for the first scan and the above-mentioned three scores for the second scan after 2 weeks. Additionally, we evaluated the scan time, knowledge test results, and pre- and post-survey scores.

### 2.7. Sample Size and Statistical Analysis

A six-person pilot study was conducted to assess the non-inferiority margin and sample size of the primary endpoint [21]. Participants in the pilot trial attended a hands-on lecture and performed ocular POCUS scans on the two SPs. Non-inferiority was defined as the condition in which the lower limit of the 95% confidence interval (95% CI) of the difference of least squares mean (lsmean) of (Group O–Group H) using the mixed model was more than a score of minus 2. The standard deviation (SD) of the total scores in Group H was 1.94 in the pilot study. Sixteen participants were required for each group to obtain 80% statistical power with a one-sided type 1 error rate of 2.5%, a non-inferiority margin of a score of 2, and a sample size ratio of 1:1. We set the sample size to 16 participants per group, assuming a dropout rate of 0%. The sample size was calculated using PASS2020. v20.0.2.

Standard descriptive statistics were used for the quantitative analysis of the collected statistical data. The ocular POCUS scan scores and scan times were analyzed using non-inferiority analysis with a generalized estimating equation model. The ONSD values were analyzed using a *t*-test, and the knowledge test scores were analyzed using a Wilcoxon rank-sum test. The scan and ONSD scores, except the total score (primary endpoint), were corrected for multiple comparisons with Bonferroni’s correction (*p*-value and CI correction). Statistical significance was set at *p* < 0.05. All missing values were excluded from the analysis. Statistical analyses were performed using SAS version 9.4 (SAS Institute, Cary, NC, USA).

## 3. Results

A total of 32 physicians, 17 EM residents, and 15 interns participated in this study. Approximately 80% of the participants rated their overall confidence in the POCUS scan as 3 or greater on the Likert scale. However, more than 60% of the participants responded that they had never performed an ocular ultrasound scan. The baseline demographics were similar between the two groups (Table 1).

### 3.1. Differences in Ocular POCUS Scores between Groups

The lsmeans (standard error, SE) of the total score for the first scan were 21.7 (0.35) in Group O and 21.3 (0.25) in Group H. The difference in the group lsmean of total scores was 0.35 (0.42), and the lower bound of the 95% CI was greater than the non-inferiority margin of minus 2 (95% CI: −0.48–1.17). For the scan and ONSD scores, the differences in group lsmean were within the non-inferiority margins, so the online group was not inferior to the hands-on group (Table 2). Table A3 shows the raw scores for each SP and ocular scan prior to adjustment.

All data were analyzed via non-inferiority analysis with a generalized estimating equation and adjusted by standardized patients and eyeballs. Endpoint data are expressed as lsmean (SE). All data except the primary endpoint (total score for the first scan) were corrected by time point of the test using multiple comparisons with Bonferroni’s correction.

### 3.2. Maintenance Evaluation of Ocular POCUS Skill after 2 Weeks

The lsmeans (SE) of the total score were 21.4 (0.22) for the first scan and 21.5 (0.26) for the second scan. The 95% CIs of the differences in the group lsmean of ocular POCUS scan scores between the first and second scans were within the range of minus 2 to 2; thus, the scores for the second scan were not statistically different from the scores for the first scan (Table 3).

All data were analyzed via equivalence test with a generalized estimating equation and adjusted by standardized patients and eyeballs. Data are expressed as lsmean (SE) and corrected by multiple comparisons with Bonferroni’s correction.

### 3.3. Time Taken to Scan

The times taken to scan ocular POCUS for each group are presented in Table A4. In the first scan, Group O took longer than Group H, but the difference was not statistically significant. There was a statistically significant reduction in scan time for the second-order scan eye (left eye of SP1) and fourth-order scan eye (left eye of SP2) compared with the first-order scan eye (right eye of SP1; Table A5. *p* < 0.001).

### 3.4. ONSD Measurement

The ONSD values measured by the participants did not differ between the groups (Table A6). In addition, the ONSD measurement accuracy did not differ significantly from the experts’ reference values for each eye (Table A7).

### 3.5. Knowledge Test and Post-Survey

As shown in Table 4, the knowledge test scores did not differ significantly between the groups. Most of the participants agreed with the need for ocular ultrasound education. More participants in Group H thought that the education method they received was satisfactory for learning ocular ultrasound (*p* < 0.001). In addition, the confidence improvement after training was higher in Group H than in Group O, but the difference was not statistically significant (*p* = 0.084).

## 4. Discussion

Ocular ultrasound is a core application of POCUS that can assist Eps in identifying various ocular diseases promptly at the bedside [7,8]. It is particularly useful when a physical examination is made difficult by the swelling of the eyelids or ocular pain or when an immediate ophthalmic backup is unavailable [7,20]. Previous studies have found that the accuracy of ocular POCUS performed by trained EPs was as high as 80% to 96%, but most of the training involved hands-on sessions [22,23]. Our study demonstrated that online training for basic ocular ultrasonography was not inferior to hands-on training. This is critical given the shift in educational priorities in the aftermath of the COVID-19 pandemic and the growing need for ultrasound education in EPs. Furthermore, it is meaningful that this study evaluated not only the level of knowledge but also the quality of the scanned images between groups.

In general, traditional POCUS education, including hands-on scanning practice, is considered more effective than online learning because ultrasound training requires a complex integration of knowledge and psychomotor skills such as practical demonstration [24]. However, this study found that the overall scores of Group O were slightly higher than those of Group H. Similar to our results, Charlotte et al. found that online learners were able to perform nerve blocks more effectively than the hands-on group (91% vs. 75% success rate, *p* = 0.17), despite being less confident and taking longer to complete the procedure [25]. Our results may have been produced because the eye is a superficially located organ and a clear scan protocol makes it easy to scan. In addition, this may have occurred because the online group could view the learning material repeatedly, concentrating on areas where they were lacking, and participants were allowed to ask questions of the investigator later if not in real time [26,27,28].

The question of how well the effectiveness of an education session is maintained is very important. This study performed a second ocular POCUS scan two weeks after the education to assess the retention of learning. As shown in Table 3, there was no statistically significant difference in scores between the first and second ocular POCUS scans. However, Kim revealed that an abdominal ultrasound workshop had varying maintenance effects on knowledge and confidence for each organ after two months to one year [29]. Since the participants in this study did not perform an ocular POCUS scan during the two-week interval, it seems that the possibility of skill improvement due to the learning effect can be excluded. However, an interval of two weeks may be insufficient to assess skill maintenance. Moreover, since they scanned the SP, it is unknown whether trained participants can properly use ocular ultrasound and reliably detect ocular pathologies in real patients. Further studies on the long-term educational effects in clinical settings are needed.

In the ED setting, the speed of examination is important. Regardless of the accuracy and usefulness of the imaging tool, its value would inevitably decrease if it took too long for the EPs to perform POCUS examination in a crowded emergency room. Except for the first eye scan, both groups took four to seven minutes to complete each eye scan, which is considered acceptable in ED settings. The first-order scan (right eye of SP1) is thought to have taken longer because participants were unfamiliar with the ocular POCUS scan and scan protocol. However, except for the third-order scan, which had to change settings such as depth and gain by applying it to a new SP, the scan time was shortened according to scan sequence. Therefore, performing basic ocular POCUS is not complex even for ocular ultrasound novices regardless of the training method. The scan time is effectively reduced even with short repetitions.

The ONSD value has been validated as an indirect assessment of intracranial pressure; therefore, the accurate measurement of ONSD now plays an essential role in trauma and medical emergency patients [5,6,30]. Several studies have shown that trained EPs were capable of accurately measuring the ONSD using bedside POCUS [6,31]. This study found no significant differences in the ONSD values between Groups O and H or between the measured and reference values (Table A5). In other words, regardless of the educational model, a single short education session can enable beginners to assess ONSD appropriately [32]. However, both groups had some erroneous ONSD measurements, with the ONSD margin measurement error (not measured on both outer parts of the optic nerve sheath) being the highest. If this point is supplemented and taught in future training, it will be possible to evaluate the increased intracranial pressure through more accurate ONSD measurements and apply it clinically.

A larger percentage of the hands-on group than of the online group reported that they were satisfied with the training method they had received. As ultrasonography is a patient-focused technique, practicing SPs may be more realistic and fulfilling. Other studies have found that participants were disappointed with a lack of hands-on instruction, feedback, and increased one-sided contact [26,28]. Recently, augmented reality and virtual reality have been used in education to improve student learning and participation. In this study, the ocular scanning process was recorded as a video with as much detail as possible and used in online education. Although no discernible difference was observed in the groups’ knowledge test scores or post-training confidence levels, low satisfaction among individuals who had no hands-on training appeared to have reached an inevitable limit of online learning. In the absence of a subjective questionnaire detailing the reasons for dissatisfaction, this study did not identify any suitable Appendix A.

This study had several limitations. First, this study focused on the effects of ultrasound education, and the participants were taught basic ocular POCUS skills in normal eyes without pathologies. Thus, it might be difficult to apply ocular POCUS to actual patients and interpret the scan results in patients with ocular pathologies. Second, although a randomized controlled trial was conducted with a statistically appropriate sample size calculated from a preliminary study, the small sample size was another limitation. In addition, four ocular POCUS scans for the two SPs were performed for each participant, which might have been insufficient to identify the statistical difference between the two groups. Third, the interval of 2 weeks was too short to evaluate skill maintenance. It is necessary to evaluate the long-term skill maintenance in future studies. Fourth, the online group could watch the learning material repeatedly. However, the total learning time did not differ significantly between the two groups. Finally, the learning material, scan protocol, and scoring system for ocular POCUS have not been validated in various settings and populations, although they were based on a review of literature.

## 5. Conclusions

This study showed that online education on basic ocular ultrasound was not inferior to hands-on education, suggesting that it could be a useful educational approach in the COVID-19 era. However, the scan protocol of ocular POCUS needs validation. Further studies are required to establish its long-term effect on trained participants’ ability to use ocular ultrasound effectively and to accurately diagnose patients.

## Figures and Tables

**Figure 1 medicina-58-00960-f001:**
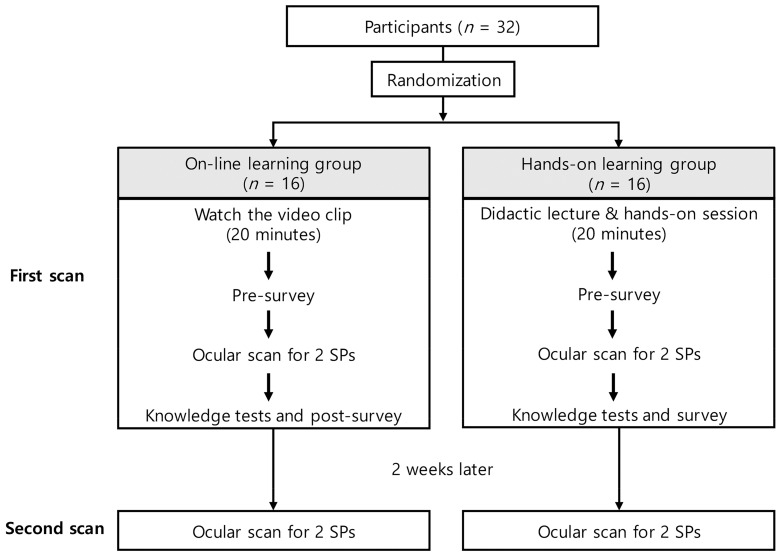
The study flow. Abbreviation: SP, standardized patient.

**Figure 2 medicina-58-00960-f002:**
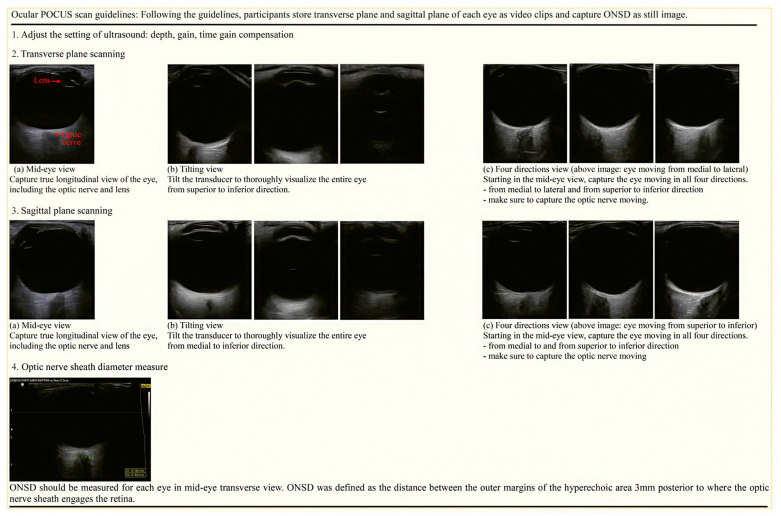
The ocular POCUS scan protocol. 1. Adjust the setting of ultrasound, 2. Transverse plane scanning, 3. Sagittal plane scanning, 4. ONSD measure. Abbreviations: POCUS, point-of-care ultrasound; ONSD, optic nerve sheath diameter.

**Table 1 medicina-58-00960-t001:** Baseline characteristics of study participants.

	Online Group (Group O, *n* = 16)	Hands-On Group (Group H, *n* = 16)	*p*-Value
Physician-grade interns/EM 1/EM 2/EM 3/EM 4	7(45)/4(25)/2(12)/2(12)/1(6)	8(50)/-/2(12)/3(19)/3(19)	0.301
Age (mean, SD)	29 (3)	30 (3)	0.283
Male	8	6	0.476
Frequency of POCUS use			
Seldom/sometimes/usually/often-always	2(12)/4(25)/3(19)/7(44)	5(31)/5(31)/1(7)/5(31)	0.662
Confidence of POCUS scan			
1/2/3/4/5 (Likert Scale)	-/4(25)/7(44)/5(31)/-	-/3(19)/7(43)/3(19)/3(19)	0.392
Number of previous ocular POCUS scan			
0/1–5/6–10/>10	11(69)/4(25)/1(6)/-	10(63)/5(31)/1(6)/-	0.499

Data are reported as *n* (%) or mean (SD). Abbreviations: SD, standardized deviation; Group O, Online group; Group H, Hands-on group; EM, emergency medicine; POCUS, point-of-care ultrasound.

**Table 2 medicina-58-00960-t002:** The non-inferiority analysis of ocular POCUS scan between groups.

	Group O	Group H	Non-Inferiority Margin (Δ)	Difference of Group Least Squares Means (95% CI) (Group O Minus Group H)	
First scan					
Total score	21.7 (0.35)	21.3 (0.25)	−2	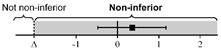	0.35 (−0.48 to 1.17)
Scan score	18.5 (0.28)	18.6 (0.27)	−2	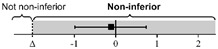	−0.13 (−1.00 to 0.74)
ONSD score	3.19 (0.14)	2.67 (0.24)	−1	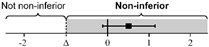	0.53 (−0.10 to 1.16)
Second scan					
Total score	21.5 (0.34)	21.0 (0.32)	−2	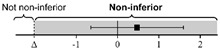	0.49 (−0.63 to 1.61)
Scan score	18.3 (0.22)	18.1 (0.23)	−2	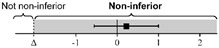	0.23 (−0.54 to 1.01)
ONSD score	3.18 (0.16)	2.99 (0.26)	−1	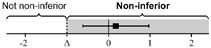	0.20 (−0.60 to 0.99)

Abbreviations: POCUS, point-of-care ultrasound; Group O, Online group; Group H, Hands-on group; CI, confidence interval; ONSD, optic nerve sheath diameter; SE, standard error.

**Table 3 medicina-58-00960-t003:** The differences in ocular POCUS scores between the first and second scans.

	First Scan	Second Scan	Difference of Group Least Squares Means (95% CI) (Second Minus First Scan)
Total score	21.4 (0.22)	21.5 (0.26)	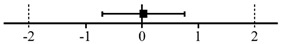	0.03 (−0.70 to 0.76)
Scan score	18.5 (0.20)	18.3 (0.12)	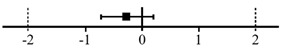	−0.26 (−0.73 to 0.21)
ONSD score	2.93 (0.14)	3.08 (0.15)	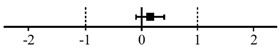	0.16 (−0.10 to 0.41)

Abbreviations: POCUS, point-of-care ultrasound; CI, confidence interval; ONSD, optic nerve sheath diameter; SE, standard error.

**Table 4 medicina-58-00960-t004:** The knowledge test and post-survey results.

	Group O	Group H	*p*-Value
Knowledge test	16 (15, 18)	17 (16, 18)	0.347
Necessity of ocular ultrasound training			
1–2/3/4–5	-/2(12)/14(88)	-/1(6)/15(94)	0.776
Subjective satisfaction with training method received			
1–2/3/4–5	1(6)/3(19)/12(75)	-/-/16(100)	<0.001 *
Adequacy of training time			
1–2/3/4–5	3(19)/5(31)/8(50)	1(6)/4(25)/11(69)	0.781
Confidence improvement after training			
1–2/3/4–5	-/3(19)/13(81)	-/2(12)/14(88)	0.084

The results were rated on a five-point Likert scale and are described as *n* (%). *p*-values are based on the Wilcoxon rank-sum test or Fisher’s exact test, and *p*-value * is significant. Abbreviations: Group O, Online group; Group H, Hands-on group.

## Data Availability

Data related to this study cannot be sent outside the hospital because of the hospital’s information security policies.

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
