# Peer review of "Online Learning versus Hands-On Learning of Basic Ocular Ultrasound Skills: A Randomized Controlled Non-Inferiority Trial"

_medicina, 2022, doi:10.3390/medicina58070960_

Round 1

Reviewer 1 Report

Firstly, thank you for opportunity to review very interested article. I don't feel qualified to judge about the English language and style due to not native language.

1. The title reflect the main subject about ocular ultrasound skills, title was clear and easy to understand.

2. The abstract summarize and reflect the work described in the manuscript.

3. The key words reflect the focus of the manuscript.

4. The manuscript adequately describe the background, present status, and significance of the study. The authors explain POCUS in the first paragraph in detail. However, I suggested the authors to explain about POCUS education in many countries which had different circumstance.

5. The manuscript describe methods in adequate detail, study subjects were clear, with demonstrate IRB number or text to human ethics consideration. This procedure in this study was ultrasound, I suggested the authors to clarify how to decrease bias from skill dependent in this study.

- line 100 was incorrected word.

6. The research objectives achieved by the experiments used in this study. 

7. The manuscript interpret the findings adequately and appropriately, highlighting the key points concisely, clearly, and logically.

8. Tables, figures, and VDO files were sufficient, good quality and appropriately illustrative of the paper contents.

9. The manuscript meet the requirements of biostatistics.

10. The manuscript cite appropriately the latest, important, and authoritative references in the introduction and discussion sections. 

Author Response

We appreciate your review and insightful comments on this study. The replies to your comments have been attached herewith.

Reviewer 2 Report

Thank you for the opportunity to review this non-inferiority assessment on the learning of ocular ultrasound skills via online learning vs. hands-on learning.

There is not much literature available regarding the topic. The sample sizes of both study groups are low and the online learning group was able to repeatedly watch the study material which might have resulted in a bias - this is, however, acknowledged by the authors in the manuscript.

The manuscript is very well written. The text is clear, easy to read, and without any obvious grammatical erros. The research methodology and the statistical analysis used are appropriate for the data collected. The results and conclusions drawn from the presented evidence are accurate.

The researchers were able to answer their initial question and concluded that online education on basic ocular ultrasound was not inferior
to hands-on education
.

I have only one minor point to improve the title of the study: please change to:

"Online learning versus hands-on learning of basic ocular ultrasound skills: A randomized controlled non-inferiority trial"

Best wishes and thank you again.

Author Response

We appreciate your review and insightful comments on this study. The reply to your comments has been attached herewith.
